# Empowering Graph Representation Learning with Paired Training and Graph Co-Attention

## Abstract

Through many recent advances in graph representation learning, performance achieved on tasks involving graph-structured data has substantially increased in recent years—mostly on tasks involving *node-level predictions*. The setup of prediction tasks over *entire graphs* (such as property prediction for a molecule, or side-effect prediction for a drug), however, proves to be more challenging, as the algorithm must combine evidence about several structurally relevant patches of the graph into a single prediction. Most prior work attempts to predict these graph-level properties while considering only one graph at a time—not allowing the learner to directly leverage structural similarities and motifs across graphs. Here we propose a setup in which a graph neural network receives *pairs* of graphs at once, and extend it with a *co-attentional* layer that allows node representations to easily exchange structural information across them. We first show that such a setup provides natural benefits on a *pairwise graph classification* task (drug-drug interaction prediction), and then expand to more generic *graph regression* and *graph classification* setups: enhancing predictions over QM9, D&D and PROTEINS, standard molecular prediction benchmarks. Our setup is flexible, powerful and makes no assumptions about the underlying dataset properties, beyond anticipating the existence of multiple training graphs.

## 1 Introduction and Related Work

We study the task of *graph-level representation learning*: i.e., computing representations of entire input graphs, for the purposes of downstream tasks (such as graph classification or regression). This is typically a step-up in complexity compared to *node classification* or *link prediction*, given that the learning algorithm must aggregate useful structural information across the graph into a single prediction—relying on only this *global* supervision signal (as opposed to having feedback from every node/edge of the graph).

Perhaps the highest challenge this kind of architecture must face is *inductivity* and *generalisation across structures*. Specifically, an inductive model must be readily applicable across several graph structures—including ones *unseen* during training. Additionally, the model is tasked with discovering interesting structural "motifs" across the entire dataset of graphs, whose presence or absence may help determine the overall predictions.

However, even enabling inductivity is not a traditionally simple task in graph representation learning, as many prior approaches (Bruna et al., 2013; Perozzi et al., 2014; Defferrard et al., 2016) are not inductive by design. Furthermore, even the models that are currently used for graph-level representation learning; e.g. Gilmer et al. (2017); Ying et al. (2018); Xu et al. (2018); Lu et al. (2019), operate over only a single graph at a time—making it challenging for them to reason about common substructures *across* graphs from a graph-level supervision signal alone.

In this manuscript, we propose the approach of **paired training**—i.e., learning representations over *pairs* of input graphs at once. Intuitively, as long as we allow for dataflow between the representations of the two graphs within a pair, this allows the graph neural network to directly observe related (sub)structures from other inputs, to solidify its decision making. We note that in the context of graph-structured inputs this may be particularly useful as, unlike simpler inputs such as images or text, there are no guarantees that different graphs within a dataset will have equal or even similar overall structure. To facilitate this dataflow, we propose the usage of **graph co-attention** for exchanging

representations of nodes across the two graphs. Intuitively, this operator performs attention (Bahdanau et al., 2014; Vaswani et al., 2017) over the fully-connected bipartite graph, with one part corresponding to all nodes in one graph. This allows every node of the first graph to detect and reuse useful patch representations in the second graph (in a form of *hierarchical graph matching*), and vice-versa.

Initially, we validate our model performance on a *pairwise graph classification task*—classifying drug pairs for side effects caused by drug-drug interactions (DDI) (Jin et al., 2017; Zitnik et al., 2018). In this setting, a pairwise approach is *natural*, as we inherently have to classify pairs of graphs. We demonstrate that learning a joint representation using graph co-attention provides substantial benefits to predictive power, setting the state-of-the-art result on this task.

From there, we demonstrate the applicability of our approach to arbitrary multi-graph datasets; for this, we leverage the QM9 dataset for predicting quantum chemistry properties of small molecules (Ramakrishnan et al., 2014). As such, it represents a challenging *graph regression* problem. We propose using *paired training* to perform regression on two molecules at once, demonstrating clear benefits to doing so. In a similar vein, we execute variants of our model on standard graph kernel classification benchmarks (Kersting et al., 2016), showing advantages to generic graph classification.

Our approach paves the way to a promising direction for graph-level prediction tasks, that is in principle applicable to any kind of multi-graph dataset, especially under availability of large quantities of labelled examples.

Our model builds up on a large existing body of work in *graph convolutional networks* (Bruna et al., 2013; Defferrard et al., 2016; Kipf & Welling, 2016a; Gilmer et al., 2017; Veličković et al., 2018), that have substantially advanced the state-of-the-art in many tasks requiring graph-structured input processing (such as the chemical representation (Gilmer et al., 2017; De Cao & Kipf, 2018; You et al., 2018) of the drugs leveraged here). Furthermore, we build up on work proposing *co-attention* (Lu et al., 2016; Deac et al., 2018) as a mechanism to allow for *individual set-structured datasets* (such as nodes in multimodal graphs) to interact. Specifically, such mechanisms have already been used for explicit matching of graph structure motifs (Li et al., 2019), and therefore represent a natural methodology for our purposes.

Overall, these (and related) techniques lie within the domain of *graph representation learning*, one of the latest major challenges of machine learning (Bronstein et al., 2017; Hamilton et al., 2017; Battaglia et al., 2018), with transformative potential across a wide spectrum of potential applications, extending outside the biochemical domain.

## 2 ARCHITECTURE

In this section, we will present the main building blocks used within our architecture for paired graph representation learning. This will span a discussion of the way the input to the model is encoded, followed by an overview of the individual computational steps of the model. As the tasks considered in this manuscript operate over molecules, we will treat all inputs as molecular graphs.

### 2.1 INPUTS

The *molecules*, $d_x$, are represented as *graphs* consisting of atoms, $a_i^{(d_x)}$ as nodes, and *bonds* between those atoms $\left(a_i^{(d_x)}, a_j^{(d_x)}\right)$ as edges. For each atom, the following input features are recorded: the *atom number*, the number of *hydrogen atoms* attached to this atom, and the *atomic charge*. For each bond, a discrete *bond type* (e.g. single, double etc.) is encoded as a learnable input edge vector, $e_{ij}^{(d_x)}$.

### 2.2 MESSAGE PASSING

Within each of the two molecules separately, our model applies a series of *message passing* (Gilmer et al., 2017) layers. Herein, nodes are allowed to send arbitrary *vector messages* to each other along the edges of the graph, and each node then aggregates all the messages sent to it.

Let $^{(d_x)}h_i^t$ denote the *features* of atom $i$ of molecule $x$, at time step $t$. Initially these are

set to projected input features, i.e.:

$$^{(d_x)}h_i^0 = f_i\left(a_i^{(d_x)}\right) \tag{1}$$

where $f_i$ is a small multilayer perceptron (MLP) neural network. In all our experiments, this MLP (and all subsequently referenced MLPs and projections) projects its input to 50 features. Note that this effectively encodes one-hot inputs such as atom type into a learnable *embedding*.

Considering atoms $i$ and $j$ of molecule $x$, connected by a bond with edge vector $e_{ij}^{(d_x)}$, we start by computing the *message*, $^{(d_x)}m_{ij}^t$, sent along the edge $j \to i$. The message takes into account both the features of node $j$ and the features of the edge between $i$ and $j$:

$$^{(d_x)}m_{ij}^t = f_e^t\left(e_{ij}^{(d_x)}\right) \odot f_v^t\left(^{(d_x)}h_j^{t-1}\right) \tag{2}$$

where $f_e^t$ and $f_v^t$ are small MLPs, and $\odot$ is elementwise vector-vector multiplication.

Afterwards, every atom $i$ *aggregates* the messages sent to it via summation. This results in an *internal message* for atom $i$, $^{(d_x)}m_i^t$:

$$^{(d_x)}m_i^t = \sum_{\forall j \in N(i)} {}^{(d_x)}m_{ij}^t \tag{3}$$

where the neighbourhood, $N(i)$, defines the set of atoms linked to $i$ by an edge.

## 2.3 CO-ATTENTION

Message passing provides a robust mechanism for encoding within-molecule representations of atoms. For learning an appropriate *joint* molecule-molecule representation, however, we allow atoms to interact *across* molecule boundaries via a *co-attentional mechanism* (Deac et al., 2018).

Consider two atoms, $i$ and $j$, of molecules $x$ and $y$, respectively. Let their features at time step $t$ be $^{(d_x)}h_i^t$ and $^{(d_y)}h_j^t$, just as before. For every such pair, we compute an *attentional coefficient*, $\alpha_{ij}^t$ using a simplified version of the Transformer (Vaswani et al., 2017) attention mechanism:

$$\alpha_{ij}^t = \text{softmax}_j\left(\left\langle \mathbf{W}_k^{t\,(d_x)}h_i^{t-1}, \mathbf{W}_k^{t\,(d_y)}h_j^{t-1}\right\rangle\right) \tag{4}$$

where $\mathbf{W}_k^t$ is a learnable *projection matrix*, $\langle \cdot, \cdot \rangle$ is the *inner product*, and the softmax is taken across *all* nodes $j$ from the second molecule. The coefficients $\alpha_{ij}^t$ may be interpreted as the *importance* of atom $j$'s features to atom $i$.

These coefficients are then used to compute an *outer message* for atom $i$ of molecule $x$, $^{(d_x)}n_i^t$, expressed as a *linear combination* (weighted by $\alpha_{ij}^t$) of all (projected) atom features from molecule $y$:

$$^{(d_x)}n_i^t = \sum_{\forall j \in d_y} \alpha_{ij}^t \cdot \mathbf{W}_v^{t\,(d_y)}h_j^{t-1} \tag{5}$$

where $\mathbf{W}_v^t$ is a learnable projection matrix.

It was shown recently that *multi-head attention* can stabilise the learning process, as well as learn information at different conceptual levels (Veličković et al., 2018; Qiu et al., 2018). As such, the mechanism of Equation 5 is independently replicated across $K$ attention *heads* (we chose $K = 8$), and the resulting vectors are *concatenated* and *MLP-transformed* to provide the final outer message for each atom.

$$^{(d_x)}n_i^t = f_o^t\left(\mathop{\Big\|}_{k=1}^K \sum_{\forall j \in d_y} {}^{(k)}\alpha_{ij}^t \cdot {}^{(k)}\mathbf{W}_v^{t\,(d_y)}h_j^{t-1}\right) \tag{6}$$

where $f_o^t$ is a small MLP and $\|$ is featurewise vector concatenation.

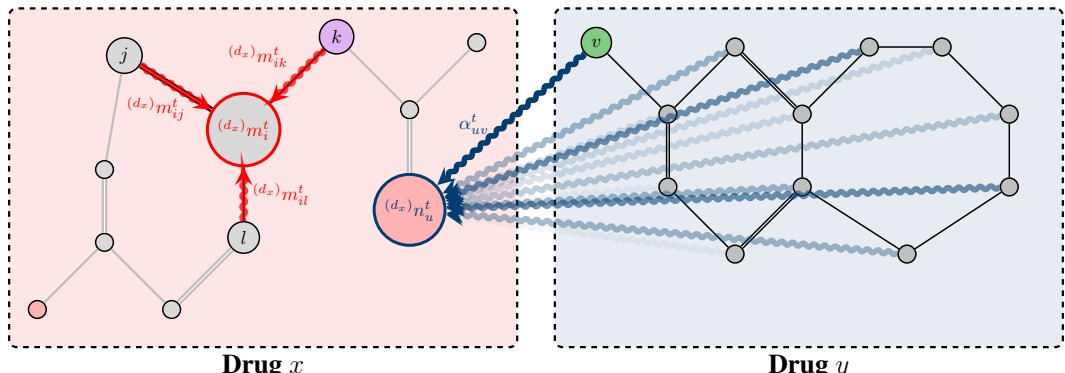

Figure 1: The illustration of a single step of message passing (computing the *inner message*, $^{(d_x)}m_i^t$), and co-attention (computing the *outer message* $^{(d_x)}n_u^t$) on two nodes ($i$ and $u$) of molecule $x$.

The computation of Equations 4–6 is replicated analogously for outer messages from molecule $x$ to atoms of molecule $y$.

We have also attempted to use other popular attention mechanisms for computing the $\alpha_{ij}^t$ values—such as the original Transformer (Vaswani et al., 2017), GAT-like (Veličković et al., 2018) and tanh (Bahdanau et al., 2014) attention—finding them to yield weaker performance than the approach outlined here.

A single step of message passing and co-attention, as used by our model, is illustrated by Figure 1.

## 2.4 UPDATE FUNCTION

Once the inner messages, $^{(d_x)}m_i^t$ (obtained through message passing), as well as the outer messages, $^{(d_x)}n_i^t$ (obtained through co-attention) are computed for every atom $i$ of each of the two molecules ($d_x/d_y$), we use them to derive the next-level features, $^{(d_x)}h_i^t$, for each atom.

At each step, this is done by aggregating (via summation) the previous features (representing a *skip connection* (He et al., 2016)), the inner messages and outer messages, followed by *layer normalisation* (Ba et al., 2016):

$$^{(d_x)}h_i^t = \text{LayerNorm}\left(^{(d_x)}h_i^{t-1} + {}^{(d_x)}m_i^t + {}^{(d_x)}n_i^t\right) \tag{7}$$

The operations of Equations 2–7 are then repeated for $T$ propagation steps—here, we set $T = 3$. Refer to Figure 2 for a complete visualisation of our architecture.

As will be demonstrated by our results, using co-attention to enable the model to propagate the information between two molecules from the beginning—thus learning a *joint* representation of the molecules—valuably contributes to the predictive power of the model.

## 2.5 READOUT AND SCORING

As we're making predictions on the level of entire molecules (or molecule-molecule pairs), we need to compress individual atom representations into molecule-level representations. In order to obtain the molecule-level vectors $d_x$, we apply a simple *summation* of its constituent atom feature vectors after the final layer (i.e. after $T$ propagation steps have been applied):

$$d_x = \sum_{\forall j \in d_x} f_r\left(^{(d_x)}h_j^T\right) \tag{8}$$

where $f_r$ is a small MLP.

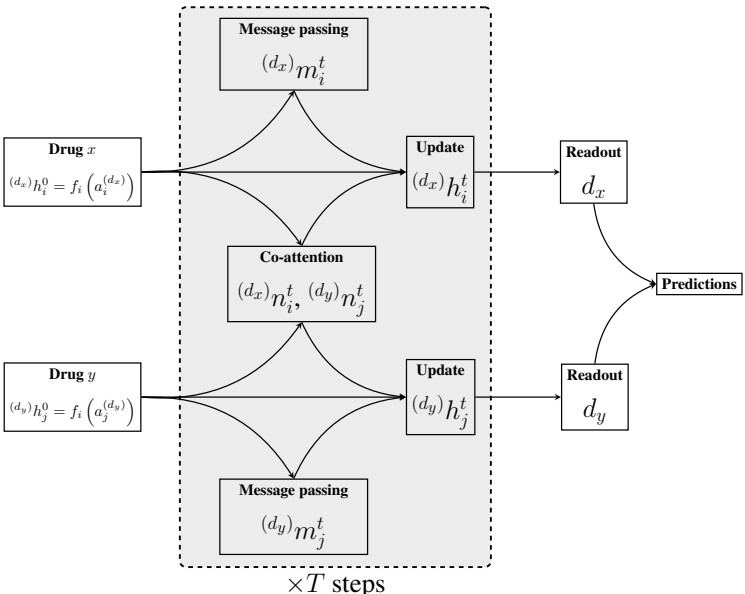

$\times T$ steps

Figure 2: A high-level overview of our paired training architecture. The next-level features of atom $i$ of molecule $x$, $^{(d_x)}h_i^t$, are derived by combining its *input features*, $^{(d_x)}h_i^{t-1}$, its *inner message*, $^{(d_x)}m_i^t$, computed using message passing, and its *outer message*, $^{(d_x)}n_i^t$, computed using co-attention over the second molecule, $d_y$.

Once the molecule vectors are computed for both $d_x$ and $d_y$, they can be used to make predictions on either $x$ and $y$ individually, or to make pairwise predictions—the exact setup varies depending on the considered task.

## 3 PAIRWISE GRAPH CLASSIFICATION: **POLYPHARMACY SIDE EFFECTS**

We now directly evaluate the benefits of our model on a *pairwise graph classification* task. This represents a natural testbed for our framework, as within the dataset, graphs are already given to us in paired form for training and evaluation.

Despite the apparent specificity of this setup, it is central to many relevant and important problems across the sciences. We study the task of *polypharmacy side effect prediction*, to be described below.

### 3.1 MOTIVATION

Diseases are often caused by complex biological processes which cannot be treated by individual drugs and thus introduce the need for concurrent use of multiple medications. Similarly, drug combinations are needed when patients suffer from multiple medical conditions. However, the downside of such treatment (referred to as *polypharmacy*) is that it increases the risk of adverse side effects, caused by the chemical-physical incompatibility of the drugs.

Uniquely within the related work in this area, our model learns a robust representation of drugs by leveraging joint information early on in the learning process. Our method outperforms previous state-of-the-art models in terms of predictive power, while using just the molecular structure of the drugs. It thus allows for the method to be applied to novel drug combinations, as well as permitting the detection to be performed at the preclinical phase.

### 3.2 DATASET AND PREPROCESSING

The drug-drug interaction data was chosen to be in agreement with the dataset used by Decagon (Zitnik et al., 2018). It is obtained by filtering the TWOSIDES side-effect dataset (Tatonetti et al.,

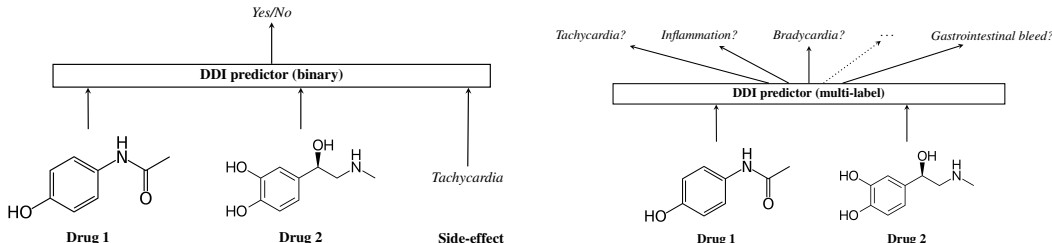

Figure 3: An overview of the binary (*left*) and multi-label (*right*) drug-drug interaction (DDI) task. In both cases, two drugs (represented as their molecular structures) are provided to the model in order to predict existence or absence of adverse interactions. For binary classification, the DDI predictor is also given a particular side effect as input, and is required to specifically predict existence or absence of it. For multi-label classification, the DDI predictor simultaneously predicts existence or absence of all side-effects under consideration.

2012), which consists of associations which cannot be clearly attributed to either drug alone (that is, those side-effects reported for any individual drug in the OFFSIDE dataset (Tatonetti et al., 2012)).

The *side effects*, $se_z$, are one-hot encoded from a set of 964 polypharmacy side-effect types that occurred in at least 500 drug pairs. The model input and loss function varies depending on whether we are performing *binary classification* for a given side effect, or *multi-label classification* for *all* side effects at once (Figure 3)—further details may be found in Appendix A. The full dataset consists of 4,576,785 positive examples. As we aim to sample a balanced number of positive and negative examples, the overall dataset size for training our model is 9,153,570 examples.

## 3.3 EXPERIMENTAL SETUP

We will refer to our model as **MHCADDI** (*multi-head co-attentive drug-drug interactions*), and **MHCADDI-ML** for multi-label classification. We perform detailed ablation studies, to stress the importance of various components of MHCADDI (especially, its co-attention mechanism). The following variations are considered:

- *MPNN-Concat*: removing co-attention, i.e. learning drug representations independently;
- *Late-Outer*: where co-attention messages are not aggregated until the last layer;
- *CADDI*: only $K = 1$ attention head.

Additionally, we compare our method with several existing strong baselines: the Multitask Dyadic Prediction (Jin et al., 2017) relying on proximal gradients and substructure fingerprints, tensor factorisation methods RESCAL (Nickel et al., 2011) and DEDICOM (Papalexakis et al., 2017), the network embedding method DeepWalk (Perozzi et al., 2014), and the multiresolution link prediction method over drug-protein interaction graphs, Decagon (Zitnik et al., 2018). We omit comparisons with traditional machine learning baselines (such as support vector machines or random forests) as prior work (Jin et al., 2017) has already found them to be significantly underperforming on this task.

## 3.4 RESULTS

We perform stratified 10-fold crossvalidation on the derived dataset. For each model, we report the area under the receiver operating characteristic (ROC) curve averaged across the 10 folds. The results of our study may be found in Table 1.

From this study, we may conclude that learning *joint* drug-drug representations by simultaneously combining internal message-passing layers and co-attention between drugs is a highly beneficial approach, consistently outperforming the strong baselines considered here (even when additional data sources are used, as in the case of Decagon). Furthermore, comparing between architectural variants, we can conclude that (further qualitative studies may be found in Appendix A):

- The fact that all our co-attentive architectures outperformed MPNN-Concat implies that it is beneficial to learn drug-drug representations *jointly* rather than separately.

Table 1: Comparative evaluation results after stratified 10-fold crossvalidation.

|  | **AUROC** |
| --- | --- |
| **Drug-Fingerprints** (Jin et al., 2017) | 0.744 |
| **RESCAL** (Nickel et al., 2011) | 0.693 |
| **DEDICOM** (Papalexakis et al., 2017) | 0.705 |
| **DeepWalk** (Perozzi et al., 2014) | 0.761 |
| **Concatenated features** (Zitnik et al., 2018) | 0.793 |
| **Decagon** (Zitnik et al., 2018) | 0.872 |
| **MPNN-Concat** (ours) | 0.661 |
| **Late-Outer** (ours) | 0.724 |
| **CADDI** (ours) | 0.778 |
| **MHCADDI** (ours) | **0.882** |
| **MHCADDI-ML** (ours) | 0.819 |

- Furthermore, the outperformance of Late-Outer by (MH)CA-DDI further demonstrates that it is useful to provide the cross-modal information *earlier* in the learning pipeline.

- Lastly, the comparative evaluation of MHCADDI against CADDI further shows the benefit of inferring *multiple mechanisms* of drug-drug interaction simultaneously, rather than anticipating only one.

We note that further relevant research in GNNs—particularly in *unsupervised learning*—could be used to further enhance our results. Given the link prediction nature of the dataset, the method of Kipf & Welling (2016b) could be useful for appropriately pre-training the GNNs used here, for example.

## 4 GRAPH REGRESSION: QUANTUM CHEMISTRY

Having confirmed that our paired training paradigm with graph co-attention yields significant returns on polypharmacy side effect prediction, we now direct attention to generalising it to *arbitrary* multi-graph datasets (even without explicitly given pairings).

We consider the challenging graph regression setup of *predicting quantum-chemical properties* of small molecules, on the QM9 dataset (Ramakrishnan et al., 2014), as previously done by e.g. Gilmer et al. (2017); Lu et al. (2019). QM9 consists of ~130,000 molecules with 12 properties for each molecule, yielding 12 regression tasks. We preprocess the dataset similarly as in prior work (by transforming the ground-truth values of the twelve quantum-chemical properties under study to the $[0, 1]$ range). We use 10,000 randomly-chosen molecules for testing and validation each; the remainder is used for training.

### 4.1 EXPERIMENTAL SETUP

We perform paired training on QM9 by pairing each input graph with $K$ different training graphs (these pairs are fixed and chosen in advance, for consistency and reproducibility purposes). We compare the following variants:

- $K = 1$ *(self)*: only pair molecules to themselves. This setup is equivalent to an MPNN (Gilmer et al., 2017) from a representational standpoint, as the co-attention layers allow for unrestricted dataflow between atoms in a molecule.

- $K = 1$ *(k-NN)*: using a pre-trained GrammarVAE (Kusner et al. (2017)) model, pair each molecule with its nearest neighbour.

- $K = 1$ *(random)*: pair each molecule to a randomly chosen training molecule.

- $K \in [2, 5]$: pair each molecule to $K - 1$ randomly chosen training molecules, and additionally to themselves.

- $K = 3$ *(k-NN)*: pair each molecule to itself and with $K - 1$ nearest neighbours obtained using pre-trained GrammarVAE.

Table 2: Mean absolute errors (MAE) over the twelve prediction tasks of QM9.

| Property | $K = 1$ (self) | $K = 1$ (k-NN) | $K = 1$ (random) | $K = 2$ | $K = 3$ | $K = 4$ | $K = 5$ | $K = 3$ (k-NN) |
|---|---|---|---|---|---|---|---|---|
| mu | 0.6267 | 1.1058 | 0.6354 | 0.6182 | 0.6242 | 0.6178 | **0.6142** | 0.6160 |
| alpha | 0.6893 | 2.2989 | 0.6913 | 0.6550 | 0.6647 | 0.6647 | **0.6523** | 0.6590 |
| HOMO | 0.0055 | 0.0140 | 0.0056 | **0.0053** | 0.0055 | 0.0054 | **0.0053** | **0.0053** |
| LUMO | 0.0057 | 0.0196 | 0.0060 | 0.0055 | 0.0055 | 0.0056 | 0.0055 | **0.0054** |
| gap | 0.0077 | 0.0242 | 0.0082 | 0.0075 | 0.0076 | 0.0076 | 0.0076 | **0.0074** |
| R2 | 59.249 | 147.25 | 61.309 | 55.563 | 57.229 | 57.341 | 55.789 | **52.823** |
| ZPVE | 0.0005 | 0.0025 | 0.0005 | 0.0005 | 0.0005 | 0.0005 | **0.0004** | **0.0004** |
| U0 | 0.2912 | 33.575 | 0.1451 | 0.1970 | 0.1902 | 0.1872 | **0.1337** | 0.2552 |
| U | 0.2980 | 29.247 | 0.1610 | 0.1557 | 0.1713 | 0.2523 | **0.1416** | 0.1984 |
| H | 0.3870 | 23.029 | 0.2587 | 0.2055 | 0.2124 | 0.2319 | **0.1363** | 0.2816 |
| G | 0.3441 | 18.554 | 0.1990 | 0.1972 | 0.1948 | 0.2574 | **0.1737** | 0.2444 |
| Cv | 0.3781 | 1.6162 | 0.3908 | 0.3665 | 0.3706 | 0.3727 | **0.3520** | 0.3534 |
| Average | 5.1902 | 21.3949 | 5.3176 | 4.8314 | 4.9731 | 4.9954 | 4.8343 | **4.6208** |

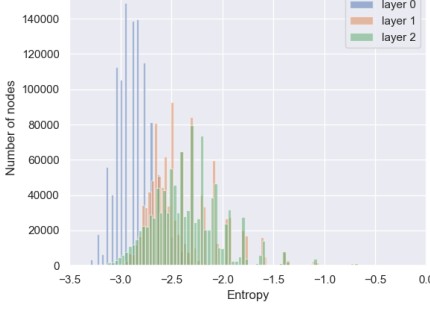 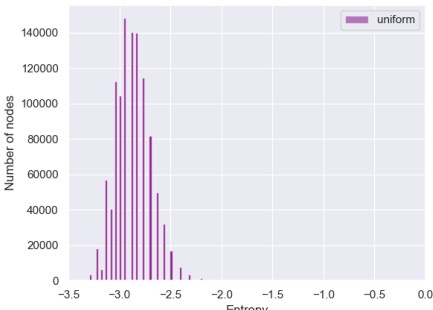

Figure 4: Histograms of entropies of co-attentional coefficients, for the three layers of a trained $K = 3$ co-attentive model (**left**) and the uniform distribution (**right**).

When training on a pair of molecules, we simultaneously optimise the mean absolute error (MAE) on both of them, for all the regression targets. At validation- or test-time, we only look at predictions for the test molecule, discarding the representation of the paired training molecule. Furthermore, whenever $K > 1$, we average the predictions across all pairings at test time.

## 4.2 RESULTS

The results on the QM9 dataset study are provided in Table 2. It may be noted that already allowing for one more random pairing on top of the self-pair ($K = 2$) yields improved performance across all of the prediction tasks. These gains remain consistent, and usually do not change much, when increasing the number of pairings beyond 2. This confirms the benefits of paired training in the graph regression setup.

Furthermore, we provide qualitative evidence for the co-attention's mechanism of action, demonstrating that it learns attentional probability distributions that are more complex than uniform aggregation—i.e. that focus on specific regions of the paired graph. We visualise, for the $K = 3$ model, the histograms of co-attention coefficient entropies across the testing dataset, and compare it against the histogram for the uniform distribution, in Figure 4. This clearly demonstrates that not only the co-attention mechanism learns complex interactions across the paired molecules, but also that it learns different mechanisms of interaction across the three co-attentive layers.

Table 3: Classification accuracy percentages after 10-fold cross-validation.

| Model | Datasets | |
|---|---|---|
| | *D&D* | *Proteins* |
| MPNN (Gilmer et al., 2017) | 70.04 | 68.15 |
| Late-Attn (ours) | 68.59 | 71.49 |
| Co-Attn (ours) | 74.11 | 71.14 |

## 5 GRAPH CLASSIFICATION

Finally, we turn our attention to the standard graph kernel classification datasets of Kersting et al. (2016). We focus on two molecular-based binary graph classification datasets: D&D and PROTEINS. In both cases, it is required to classify protein structures based on whether they are enzymes. D&D comprises 1,178 graphs of protein structures (with 284 nodes and 715 edges on average). PROTEINS comprises 1,113 graphs (with 39 nodes and 72 edges on average).

For both datasets, we train a co-attentive GNN with propagation rules as described in the Architecture section. The GNN consists of three interleaved layers of message passing and co-attention, computing 64 features each. As is common-practice for these datasets (see e.g. Ying et al. (2018)), we provide the accuracy results on them after 10-fold crossvalidation. For paired training purposes, we select a random training graph to pair with for each batch, and otherwise proceed as described for QM9. Random pairing was sufficient to demonstrate clear gains, and thus we leave more complex pairings for future analysis.

For evaluation purposes, we compare our co-attentive GNN with paired training against a version without co-attention (referred to as *MPNN*), as well as a version with only one co-attentive layer at the end (referred to as *Late-Attn*).

The results of this study may be found in Table 3. Evidently, the co-attentive method improves over its corresponding MPNN baseline on both datasets. Furthermore, on D&D, it outperforms the late-applied co-attention, providing evidence of useful hierarchical structure matching between the two paired graphs.

## 6 CONCLUSIONS

We have presented a novel way of training graph neural networks on multi-graph datasets, relying on making predictions *jointly*, in pairs of graphs—the **paired training** approach. Additionally, we allowed for arbitrary representation exchange between these graphs by way of **co-attentive mechanisms**. The two combined allow for extraction of stronger and more robust representations as opposed to single-graph learning, which is a claim we verified across several established molecular prediction tasks: polypharmacy side effect prediction (where we set a state-of-the-art result), quantum chemistry properties prediction and graph classification. As a flexible and generic approach which doesn't rely on dataset properties in any way, so long as it consists of multiple graphs, we believe it to be a useful direction to explore for graph representation learning as a whole.

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

## A  POLYPHARMACY SIDE EFFECT TASK DETAILS

### A.1  DATASET GENERATION

- For binary classification (Figure 3 (Left)), the input to our model is a *triplet* of two drugs and a side effect $(d_x, d_y, se_z)$, requiring a binary decision on whether drugs $x$ and $y$ adversely interact to cause side effect $z$.

- For multi-label classification (Figure 3 (Right)), the input to our model is a pair of two drugs $(d_x, d_y)$, requiring 964 *simultaneous* binary decisions on whether drugs $x$ and $y$ adversely interact to cause *each* of the considered side effects. Note that, in terms of learning pressure, this model requires more robust *joint representations* of pairs of drugs—as they need to be useful for all side-effect predictions *at once*.

In order to compensate for the fact that TWOSIDES contains only positive samples, appropriate *negative sampling* is performed for the binary classification task:

- During training, tuples $(\tilde{d}_x, \tilde{d}_y, se_z)$, where $\tilde{d}_x$ and $se_z$ are chosen from the dataset and $\tilde{d}_y$ is chosen at random from the set of drugs different from $d_y$ in the true samples $(\tilde{d}_x, d_y, se_z)$.
- During validation and testing, we simply randomly sample two distinct drugs which do not appear in the positive dataset.

## A.2 TRAINING SETUP

To further regularise the model, dropout (Srivastava et al., 2014) with $p = 0.2$ is applied to the output of every intermediate layer. The model is initialised using Xavier initialisation (Glorot & Bengio, 2010) and trained using the Adam SGD optimiser (Kingma & Ba, 2014), with early stopping. The learning rate after $t$ iterations, $\eta_t$, is derived using an exponentially decaying schedule:

$$\eta_t = 0.001 \cdot 0.96^{t \cdot 10^{-6}} \tag{9}$$

## A.3 BINARY CLASSIFICATION

In the binary classification case, the side effect vector $se_z$ is provided as input. We then leverage a scoring function $f$ similar to the one used by Yoon *et al.* Yoon et al. (2016) to express the likelihood of this side effect occuring:

$$f(d_x, d_y, se_z) = \left\|\mathbf{M}_h d_x + se_z - \mathbf{M}_t d_y\right\|_2^2 + \left\|\mathbf{M}_h d_y + se_z - \mathbf{M}_t d_x\right\|_2^2 \tag{10}$$

where $\|\cdot\|_2$ is the $L_2$ norm, and $\mathbf{M}_h$ and $\mathbf{M}_t$ represent the *head node* and *tail node* space mapping matrices, respectively.

The model is then trained end-to-end with gradient descent to optimise a *margin-based* ranking loss:

$$\mathcal{L} = \sum_{d_x, d_y, se_z} \sum_{\tilde{d}_x, \tilde{d}_y, se_z} \max(0, \gamma - f(d_x, d_y, se_z) - f(\tilde{d}_x, \tilde{d}_y, se_z))) \tag{11}$$

where $(d_x, d_y)$ is a drug-drug pair exhibiting side effect $se_z$, and $(\tilde{d}_x, \tilde{d}_y)$ is a drug-drug pair not exhibiting it. $\gamma > 0$ is the *margin* hyperparameter.

## A.4 MULTI-LABEL CLASSIFICATION

Here, all side effects are predicted simultaneously, and accordingly we define a *prediction layer*, parametrised by a learnable weight matrix $\mathbf{W}_p$ and bias $b_p$. This layer consumes the concatenation of the two drug vectors and projects it into a score for each of the 964 side effects. These scores are then converted into probabilities, $y_{xy}^z$, of each side effect, $z$, occurring between drugs $x$ and $y$ using the *logistic sigmoid* nonlinearity, $\sigma$, applied elementwise:

$$y_{xy}^z = \sigma \left(\mathbf{W}_p[d_x \| d_y] + b_p\right)_z \tag{12}$$

We can then once again train the model end-to-end with gradient descent, using *binary cross-entropy* against the ground truth value, $\hat{y}_{xy}^z$ (which is a binary label indicating whether side effect $z$ actually occurs between drugs $x$ and $y$). The loss function is as follows:

$$\mathcal{L}_{BCE} = \sum_{d_x, d_y, z} \hat{y}_{xy}^z \log y_{xy}^z + \left(1 - \hat{y}_{xy}^z\right) \log \left(1 - y_{xy}^z\right) \tag{13}$$

## A.5 QUALITATIVE RESULTS

The effectiveness of the learnt joint representations may be investigated *qualitatively* as well, and to that end, we have devised a controlled visualisation experiment. Our objective is to investigate the distribution of learnt drug-drug embeddings with respect to individual side effects.

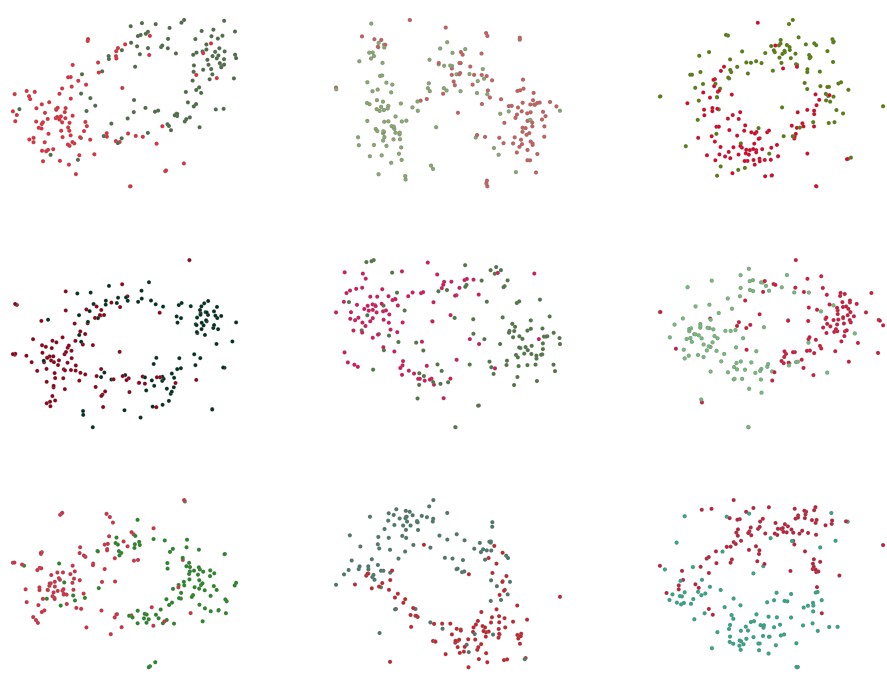

Figure 5: t-SNE projections of 200 learnt drug-drug representations, across 9 side-effects. Each side effect is colour-coded with a shade of red for positive drug-drug pairs and a shade of green for negative pairs.

We start with a pre-trained MHCADDI model. For each side effect in a sample of 10, we have randomly sampled 100 drug-drug pairs exhibiting this side effect, and 100 drug-drug pairs not exhibiting it (both unseen by the model during training). For these pairs, we derived their embeddings, and projected them into two dimensions using t-SNE Maaten & Hinton (2008).

The visualised embeddings may be seen in Figure 5 for individual side-effects, and Figure 6 for a combined plot. These plots demonstrate that, across a single side-effect, there is a discernible clustering in the projected 2D space of drug-drug interactions. Furthermore, the combined plot demonstrates that strongly side-effect inducing drug-drug pairs tend to be clustered together, and away from the pairs that do not induce side-effects.

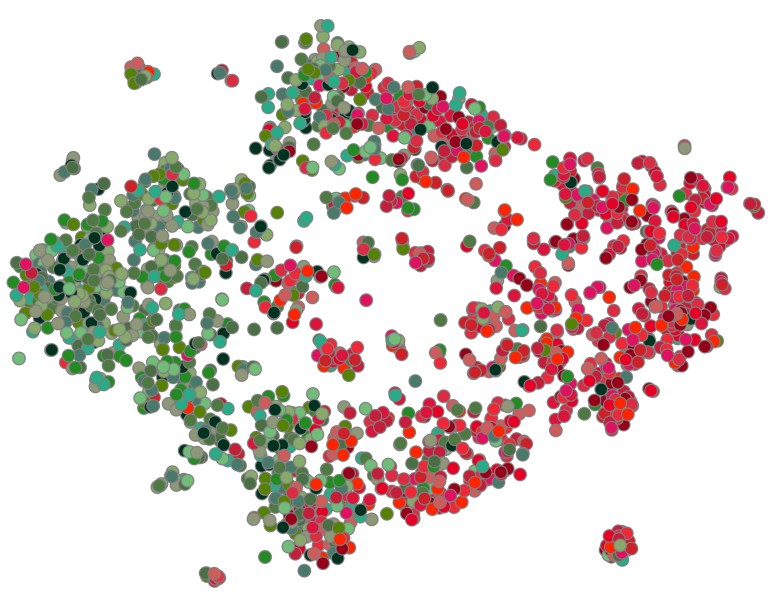

Figure 6: t-SNE projection of 2000 learnt drug-drug representations, sampled to cover 10 side effects. Each side effect is colour-coded with a shade of red for positive drug-drug pairs and a shade of green for negative drug-drug pairs.

