# OpenReview forum: "Empowering Graph Representation Learning with Paired Training and Graph Co-Attention"
_ICLR.cc/2020/Conference — Reject_

### Official Review · AnonReviewer1 · 2019-10-21
**Official Blind Review #1**

**Rating:** 3

**Review:**

This work injects a multi-head co-attention mechanism in GCN that allows one drug to attends to another drug during drug side effect prediction. The motivation is good with limited technical novelty. The paper is well-written and well organized.


For MHCADDI, it is performing binary classification for all side effect labels. It is different from Decagon’s setting, hence not comparable. Maybe also include Decagon-Binary?


Missing baseline: as its main innovation is using co-attention, it should compare with concatenated embedding generated from Graph Attention Network so that we know co-attention is better than independent attention on each drug (seems the authors have already attempted to do so but did not report it). Current baselines such as Decagon only use GCN with no attention mechanism. It could be also benefited by including VGAE.

**Experience Assessment:**

I have published in this field for several years.

**Review Assessment: Checking Correctness Of Derivations And Theory:**

I did not assess the derivations or theory.

**Review Assessment: Checking Correctness Of Experiments:**

I carefully checked the experiments.

**Review Assessment: Thoroughness In Paper Reading:**

I read the paper thoroughly.

---

> ### Author Response · Authors · 2019-11-15
> **Reply to AnonReviewer1**
>
> Firstly, we would like to thank the reviewer for their kind thoughts and comments on our paper!
>
> Regarding your comment on the technical novelty: our proposal concerns the combination of paired training with co-attention, which seeks to generically exploit and match similarities between two given graph structures (using a co-attention mechanism) in a hierarchical way (through stacking of co-attentive layers).
>
> This is a generic idea that extends beyond just the drug-drug side effect prediction task, and we had demonstrated its utility with further graph regression experiments on QM9. In the revision of the paper we submitted just now, we also include results on two standard graph classification datasets (D&D/PROTEINS; Section 5). In both of these kinds of datasets, it can be observed that no known way of “pairing” the graph structures is given, yet our methodology manages to extract additional benefits compared to its respective baseline GNN.
>
> Thank you for your comments regarding existing baselines on the drug-drug interaction task. We would like to note that the proposed comparison with concatenated GAT embeddings is already present in the paper (and given under the name of “MPNN-Concat”). This architecture has turned off co-attention, and comparatively evaluating it against the full co-attentive model (along with additional ablation studies against models such as CADDI and Late-Outer) represent our key evaluation, directly demonstrating the benefits of the different components of our model.
>
> With respect to this, our comparison with Decagon primarily serves to put our results into context with the existing state-of-the-art (i.e. to show that our results are competitive). It should also be highlighted that our method does not require additional information that Decagon uses (such as protein-protein interaction graphs).
>
> Lastly, thank you for pointing VGAEs as a possible additional option -- we have now cited VGAE appropriately at the end of Section 3.
>
> We thank you once again for your thoughtful review!

---

### Official Review · AnonReviewer3 · 2019-10-22
**Official Blind Review #3**

**Rating:** 3

**Review:**

In this paper, the authors proposed a method to extend graph-based learning with a co-attentional layer. Feeding graphs pairwisely into the model allows nodes to easily exchange information with nodes in other graphs as well as within the graph. This method outperforms other previous ones on a pairwise graph classification task (drug-drug interaction prediction).
This model is generalized from Neural Message Passing for Quantum Chemistry (Justin Gilmer et al.) and Graph Attention Networks (Petar Velickovic et al.), but most ideas are directly from the two previous papers. Combining the two methods do provide insights into understanding the interactions between graphs and get really good results on DDI prediction, but the novelty is limited.
Questions:
1 Are atoms encoded as only atom numbers, charges and connected hydrogen atoms? Because some atoms might have much larger atom numbers than others, e.g. carbon (6) and sulfur (16), will there be some scale problems? Will one-hot encoding of atom type help (like in Neural Message Passing for Quantum Chemistry)?
2 According to the paper, bond types will be encoded as e_{ij}. But in molecules, bond type is way more complex than only single/double/triple bonds, especially for drug molecules which are enriched for aromatic systems. For example, bonds in benzene or pyridine rings are between single and double (also not necessarily 3/2). Are there other possible methods to encode graph edges?
3 In result table 2 of Section 4 (quantum chemistry), I didn’t see a principle of choosing K value and choosing neighbors because different properties reaches the lowest MAE at different K values. This might cause some confusion in real application. Moreover, the authors should compare the performance with previous methods.


**Experience Assessment:**

I have read many papers in this area.

**Review Assessment: Checking Correctness Of Derivations And Theory:**

I assessed the sensibility of the derivations and theory.

**Review Assessment: Checking Correctness Of Experiments:**

I assessed the sensibility of the experiments.

**Review Assessment: Thoroughness In Paper Reading:**

I read the paper at least twice and used my best judgement in assessing the paper.

---

> ### Author Response · Authors · 2019-11-15
> **Reply to AnonReviewer3**
>
> We would like to start by thanking the reviewer for their highly constructive and useful comments.
>
> Initially, we would like to address your comment on the technical novelty: our proposal concerns the combination of paired training with co-attention, which seeks to generically exploit and match similarities between two given graph structures (using a co-attention mechanism) in a hierarchical way (through stacking of co-attentive layers).
>
> This is a generic idea that extends beyond just the drug-drug side effect prediction task, and we had demonstrated its utility with further graph regression experiments on QM9. In the revision of the paper we submitted just now, we also include results on two standard graph classification datasets (D&D/PROTEINS; Section 5). In both of these kinds of datasets, it can be observed that no known way of “pairing” the graph structures is given, yet our methodology manages to extract additional benefits compared to its respective baseline GNN.
>
> To answer your questions in turn (we are very open to further discussions, of course):
>
> 1. We would like to confirm that, indeed, we do leverage one-hot encodings of the atom type (as is common practice in the MPNN paper, for example). Effectively, a separate vector representation is learnt for each atom in this way. We found this to be a critical component to the performances we obtained on the computational chemistry datasets. Thank you for pointing this out to us -- we are now making it more clear in the (just submitted) revision of the paper.
>
> 2. Currently, we seek to encode all bond types as discrete (one-hot) representations. The approach is typically to assume “special” edge types for bonds that are situated in e.g. a benzene ring, (to specify that they are not quite e.g. single or double bonds).
>
> 3.
> a) We provided some loose guidelines on choosing K in the paper -- generally, choosing any K > 1 will yield benefits compared to self-pairing and other K = 1 variants. From there, the differences are more fine-grained and depend on the chemical property being predicted---but the overall performance does not change drastically.
>
> b) We would like to note that we have now updated the QM9 prediction MAEs to match unscaled values. Unfortunately, a direct comparison of our work with previously published numbers is extremely difficult, given that different manuscripts utilise different scales of the output labels (e.g. Gilmer et al. incorporate a ratio to the DFT chemical accuracy). In addition, our architecture aims to predict all molecular properties simultaneously, which is not comparable to training individual models for every property (as done by Gilmer et al., for example). With this in mind, we note that the (K = 1, self) model is, in terms of data flow between the atom representations, equivalent to the state-of-the-art MPNN model of Gilmer et al.
>
> Thank you very much for your review, which has certainly helped make our contributions stronger!

---

### Official Review · AnonReviewer4 · 2019-10-31
**Official Blind Review #4**

**Rating:** 3

**Review:**

The paper presents a model to classify pairs of graphs which is used to predict sided effects caused by drug-drug-interactions (DDI).

The contribution of this work is to add attention connections between two graphs such that each node operation from one graph can attend on the nodes of the other graph. The paper shows good results in DDI prediction, although the performance gap with previous works (Zitnik et al., 2018) is modest.

In the related work they mention some works from Graph Neural Networks literature. But works from the benchmark experiments are not explained. I think they could also explain which are the similarities and differences of the proposed method vs these works they are comparing to.

Another way of improving the paper could be running more experiments beyond the QM9 dataset to corroborate the good performance of the algorithm.

In equation (2), a message that goes from node “j” to node “i” does not include node “i” as input into the edge operation. I think the GNN would be more powerful if both nodes "i" and "j" are input into the edge operation.

In summary, the main contribution of the paper is to add attention connections between two graphs. I do not feel it is innovative enough.


**Experience Assessment:**

I have published one or two papers in this area.

**Review Assessment: Checking Correctness Of Derivations And Theory:**

I assessed the sensibility of the derivations and theory.

**Review Assessment: Checking Correctness Of Experiments:**

I assessed the sensibility of the experiments.

**Review Assessment: Thoroughness In Paper Reading:**

I read the paper at least twice and used my best judgement in assessing the paper.

---

> ### Author Response · Authors · 2019-11-15
> **Reply to AnonReviewer4**
>
> We would like to thank the reviewer for their careful and detailed review.
>
> Initially, we would like to address your comment on the innovativeness of our contribution: our proposal concerns the combination of paired training with co-attention, which seeks to generically exploit and match similarities between two given graph structures (using a co-attention mechanism) in a hierarchical way (through stacking of co-attentive layers).
>
> This is a generic idea that extends beyond just the drug-drug side effect prediction task, and we had demonstrated its utility with further graph regression experiments on QM9. In the revision of the paper we submitted just now, as per your advice, we also include results on two standard graph classification datasets (D&D/PROTEINS; Section 5). In both of these kinds of datasets, it can be observed that no known way of “pairing” the graph structures is given, yet our methodology manages to extract additional benefits compared to its respective baseline GNN.
>
> Regarding our DDI results, we note that our primary evaluation is the comparative ablation of various aspects of the MHCADDI model (directly evaluating the benefits of individual decisions such as paired training, co-attention, and multi-head attention, against a strong GNN-based baseline). We include results from Decagon (and others) to situate our results against the existing state-of-the-art, demonstrating the method is in essence competitive. On the issue of the “modest” gains compared to Decagon, we would like to highlight that our method does not require additional information that Decagon uses (such as protein-protein interaction graphs).
>
> We agree with the comment about the related work on DDI, and have now expanded each of the proposed strong baselines (RESCAL, DEDICOM, DeepWalk and Decagon) in turn within Section 3.3.
>
> Finally, we would like to note that we have attempted a variant of our model that takes into account both the sender and receiver node (i and j) when computing edge messages. We have found no tangible benefits to this implementation for the datasets considered.
>
> Your review has been very valuable to us in terms of improving the paper -- thank you once again!

---

### Decision · Program_Chairs · 2019-12-19

**Decision:**

Reject

**Comment:**

The paper proposes combining paired attention with co-attention. The reviewers have remarked that the paper is will written and that the experiments provide some new insights into this combination. Initially, some additional experiments were proposed, which were addressed by the authors in the rebuttal and the new version of the paper. However, ICLR is becoming a very competitive conference where novelty is an important criteria for acceptance, and unfortunately the paper was considered to lack the novelty to be presented at ICLR.